# *Mucilaginibacter* sp. Strain Metal(loid) and Antibiotic Resistance Isolated from Estuarine Soil Contaminated Mine Tailing from the Fundão Dam

**DOI:** 10.3390/genes13020174

**Published:** 2022-01-19

**Authors:** Ana L. S. Vasconcelos, Fernando Dini Andreote, Thaiane Defalco, Endrews Delbaje, Leticia Barrientos, Armando C. F. Dias, Fabricio Angelo Gabriel, Angelo F. Bernardino, Kattia Núñez-Montero

**Affiliations:** 1College of Agriculture “Luiz de Queiroz”, Universidade de São Paulo, Piracicaba 13418-900, Brazil; thaiecology@gmail.com (F.D.A.); thai.ecology@gmail.com (T.D.); dias147@gmail.com (A.C.F.D.); k.nunez03@ufromail.cl (K.N.-M.); 2Center Nuclear Energy in Agriculture, Universidade de São Paulo, Piracicaba 13416-000, Brazil; endrews.delbaje@usp.br; 3Laboratory of Molecular Applied Biology, Centro de Excelencia em Medicina Traslacional, Universidad de La Frontera, Temuco 01145, Chile; Leticia.barrientos@ufronteira.cl; 4Department of Oceanografy, Universidade Federal do Espírito Santo, Vitória 29075-910, Brazil; fabricio.gabriel@outlook.com (F.A.G.); bernardino.ufes@gmail.com (A.F.B.); 5Centro de Investigación em Biotecnología, Instituto Tecnológico de Costa Rica, Cartago 30101, Costa Rica

**Keywords:** Bacteroidetes, heavy metal, EPS, genomic island, SAMARCO disaster

## Abstract

In 2015 a mine dam with Mn-Fe-rich tailings collapsed releasing million tons of sediments over an estuary, in the Southwest of Brazil. The tailings have a high concentration of metals that contaminated soil until the present day. The high contaminant concentrations possibly caused a selection for microorganisms able to strive in such harsh conditions. Here, we isolated metal(loid) and anti-biotic resistance bacteria from the contaminated estuarine soil. After 16S rDNA sequencing to identify the strains, we selected the *Mucilaginibacter* sp. strain for a whole-genome sequence due to the bioprospective potential of the genus and the high resistance profile. We obtained a complete genome and a genome-guided characterization. Our finding suggests that the 21p strain is possibly a new species of the genus. The species presented genes for resistance for metals (i.e., As, Zn, Co, Cd, and Mn) beyond resistance and cross-resistance for antibiotics (i.e., quinolone, aminoglycoside, β-lactamase, sulphonamide, tetracycline). The *Mucilaginibacter* sp. 21p description as new species should be further explored, as their extracellular polymeric substances and the potential of this strain as bioremediation and as a growth promoter in high met-al(loid) contaminated soil.

## 1. Introduction

Metal pollution is one of the emerging environmental contamination problems, mainly due to anthropogenic activities such as mine dam collapses [1]. One of the largest mine tailing disasters occurred in Brazil in 2015. The Fundão Dam spilled millions of tons of tailing on the Doce river basin, which later reached its estuary region enriching with exogenous metal(loid) [2]. The presence of tailing is still a problem for the population in 39 Brazilian cities [3]. The metal(loid) contamination in the estuary area is even higher due to the soil redox process, which may facilitate metal remobilization to the aquatic biota [4]. Long-term monitoring of this ecosystem revealed soils with an elevated concentration of As, Cd, Pb, Zn, and Mn sediments and water beyond the threshold, causing cumulative effects on fishes and along their other trophic levels [5].

Investigating sediment bacteria is an opportunity to understand resistance/tolerance adaptation and molecular mechanisms involved, with important applications in the field of bioremediation and biostimulation [6]. Metals exert selective pressure on microbial communities, driving processes in the evolution of metal resistance determinants [7]. Indigenous bacteria isolated from the indigenous group of bacteria from the contaminated area is usually the best cost-effective, eco-friendly bioremediation technology. One mechanism of metal resistance is the production of extracellular polymeric substances (EPS). Some bioremediation bacteria were described absorbing metalloids (e.g., Cd, Cu) and increasing plants growth due to EPS production [8].

*Mucilaginibacter* is a recently described bacterial genus (named in 2008), characterized by species with a high capacity for EPS production [9]. The genus was described as encompassing bacteria that play a key role in precipitating metal and growth-promoting in plants in contaminated and stressed environments. Beyond this, components of *Mucilaginibacter* have a versatile metabolic characteristic allowing a broad ecological niche and habits as lakes [10,11], marine sand [12], phyllosphere [13], mine soil [14], wood [15], straw [16], etc., what is favorable for it use as bioremediation.

In this study, we aimed to isolate and identify indigenous bacteria with resistance to metal(loid) in estuarine soils and explore indigenous bacteria’s resistance mechanism for metal(loid) exposition through the genome. We hypothesized that metal(loid) contamination could increase the resistance genes of antibiotics and metals by co-selection.

## 2. Materials and Methods

### 2.1. Site Description and Sampling

The Rio Doce is a major river in southeastern Brazil. The basin has been modified for several decades of human activities, e.g., waste discharge, agriculture, aquaculture, industrial and mining activities. In November 2015, a large amount of Fe-enriched mine tailings mostly composed by highly crystalline Fe oxyhydroxides (e.g., hematite and goethite) were dumped into Rio Doce Basin after the Fundão Dam collapse, reaching the Rio Doce Estuary. The presence of tailing was visible in the water and in first 10 cm of the soil.

For this study, contaminated soil was collected in December 2018 from the Rio Doce estuary on the southwest coast of Brazil (latitude 19°23′28″ S, longitude 40°04′20″ O), 24 months after the disaster, (Figure 1). The soils were sampled using polyvinyl chloride tubes attached to a sampler used flooded soils. After the tubes were hermetically sealed and transported in a vertical position (at approximately 4 °C). We collected the first 5–10 cm depth from the core, where mine tailing was visible and homogenized in Ziplock bag and kept in cold stored chamber (5 °C) for 5 days. Soil redox potentials (Eh) were measured during sample collection using a Pt electrode, with the Eh values adjusted to a calomel reference electrode by adding +244 mV. Soil pH values were recorded with a glass electrode, previously calibrated with standard solutions of pH 4.0 and 7.0. A sub-sample was collected for total metal contents analysis, determined by plasma atomic emission spectroscopy (ICP-OES) after microwave-assisted triacid digestion (HCl + HNO_3_ + HF; USEPA, Washington, DC, USA, 1996).

### 2.2. Isolation and Inhibitory Tests

In the laboratory, we add 100 mL of sterile water to 5 g of soil and incubated at 30 °C with 150 rpm agitation for 6 h. The bacterial isolation was performed by spread plate technique using soil sample dilutions in 10% Tryptic Soy Agar (Merck) solid media with 1 mg/mL of nystatin and Mn (1.6 mg/mL) [17]. The TSB agar was made up of 10% of the recommended concentration normally used to create a low nutrient media simulating the low nutrients in the water samples [18]. The plates were incubated for 72 h at 35 °C. The colonies that survived in this concentration of Mn, were reselected for further metal resistance test including Zn (500 mg/mL); Cd (150 mg/mL); Co (800 mg/mL). Due to the high frequency of cross-resistance among metals and antibiotics, we also conducted antibiotic resistance test for the 21 strains isolated using five class of antibiotic: β-lactam (ampicillin 10, 200, 300 µg/mL), aminoglycosides (kanamycin 50, 100 µg/mL, streptomycin 20, 50, 100 µg/mL, tetracycline 10, 30, 50 µg/mL, amphenicols (chloramphenicol 10, 50, 70 µg/mL) quinolones (nalidixic acid 50, 100, 200 µg/mL) [19,20]. Metals and antibiotics solutions were added to the medium after sterilization by filtering using a 0.22 m membrane (Millipore nitrocellulose GSWP 04700) [17]. All the solution were added in warm media avoiding precipitation. The antibiotics and metal were diluted in sterile water, except for chloramphenicol which was diluted in ethanol 70 %. All 21 isolates were inoculated in the plate and the ones that presented minimum grown were considered resistant. After the tests we stored in tryptic soy broth (TSB; Fisher Scientific Inc., Hampton, NH) with 20 % glycerol at −80 °C.

### 2.3. DNA Extraction and 16S rDNA Sequencing

A colony of each isolate was cultured in tryptic soy broth incubated at 30 °C with 150 rpm agitation for 24 h. The DNA was extracted using the Fenol-Chloroform method [18]. The 16S rDNA was amplified using the PCR method with primers 1492R (TACGGYTACCTTGTTACGACT) 27F (GAGAGTTTGATCCTGGCTCA) [20]. The PCR products were purified with Charge Switch™ PCR Clean-Up Kit (Invitrogen™, Carlsbad, CA, USA) then sequenced and ran on the ABI 3730XL capillary DNA. A contiguous sequence was constructed with forward and reverse sequencing data resulting in a fragment of approximately 900 bp, with DNA Baser Sequence Assembler v4 (2013) (Heracle BioSoft, Arges, Romania).

### 2.4. Bioinformatic Analysis of 16S and Phylogeny

The sequences were trimmed (phred > 20) by Codoncode Aligner v. 2.0.4 and SeaView v.4. The sequences were aligned using MAFFT version 7 with L-INS-I toll [21]. The highest similarities (as percentage similarities) and accession numbers are given in Appendix A. Sequences were compared with those in the NCBI database (GenBank) using the BLAST Sequence Analysis Tool (BLASTN). Isolates with a >99% match to the published sequences were identified to the species level, and those with a >97% match was identified to the genus level. For phylogenetic tree construction we used IQ-tree [22] and MEGA 7 [23]. Tree visualization was performed with ITOL [24].

### 2.5. DNA Extraction and Whole Genome Sequencing of Mucilaginibacter sp.

The selected isolated bacterium (strain 21p) was incubated in 40 mL of liquid tryptic soy broth 10% for 48 h, 30 °C and 400 rpm (SK-O330). The media was centrifuged until 1 mL of bacteria pellet was obtained. DNA was extracted with Wizard^®^ Genomic DNA Purification System (Promega, Leiden, The Netherlands) using the manufacturer’s protocol. DNA integrity was confirmed in the electrophoreses gel. DNA quantity and quality were assessed by fluorometry using a QuantiFluor^®^ ONE dsDNA System (Promega Corporation, Fitchburg, WI, USA) and ratios 260/280 and 260/230 absorbance by spectrophotometry.

The genome was sequenced using Illumina, for high quality short reads sequencing and MinION, for long reads to achieve a complete genomic sequence. For Illumina, the genomic DNA library was constructed using a Nextera XT library prep kit (Illumina, Inc., San Diego, CA, USA) with paired-end reads (2 × 150 bp) on a MiSeq v3 platform 1GBps (Illumina, San Diego, CA, USA). For MiniON sequencing, libraries were prepared with Rapid Barcoding Sequencing QK-RBK004 (Oxford Nanopore Technologies [ONT], Oxford, UK) platform using the MinKNOW software, Version 4.5.0 (Appendix A).

### 2.6. Whole Genome Hybrid Assembly

The bioinformatic workflow of the hybrid, MinION and MiSeq assemblages (Appendix A) Illumina sequencing data were converted to fastq format using the MiSeq reports program, producing 14,288,200 raw reads. To perform quality trimming (phred < 20) and adapter removal, pre-processing was carried out with the fastp tool [25]. Besides, for long ONT-reads fastp data files were obtained using Guppy base-calling v.3.6.0 (https://staff.aist.go.jp/yutaka.ueno/guppy/, accessed on 20 October 2020). The mean read quality of the raw long reads was scored using NanoPlot 1.0.0 and trimmed with NanoFilt and Porechop [26]. Trimmed ONT and Illumina reads were then de novo assembled with the hybrid assembly method in the Unicycler 0.4.8 pipeline, which functions mainly as an optimizer of SPAdes 3.13.1 [27], but includes the long ONT sequences to complete the assembly. The genome was deposit in GenBank, SUB8286515, under the BioProject accession number PRJNA667924.

### 2.7. Taxonomic Affiliation and Phylogenetic Analysis

Whole genome comparison was performed through Average Nucleotide Identity using BLAST algorithm (ANIb) [28]. Two comparisons were made one with the 32 *Mucilaginibacter* representative sequence available at NCBI—excluding partial and anomalous results and the other with 60 complete assembly sequences deposited in NCBI using OrthoANI v1.4 [29].

Furthermore, phylogenetic analysis was performed based on 120 single-copy conserved marker proteins (amino acid sequence) from bacteria, selected and aligned with GTDB-Tk v0.3.2 based on 94,759 bacterial genomes [30]. Maximum likelihood phylogenomic tree was constructed with RAxML v8.0.0 [21] with 1000 bootstraps using the PROTGAMMAIGTR model. IQ-TREE web server applied Akaike information criterion (Kalyaanamoorthy) to find the most accurate substitution model method using the following parameters: GTR substitution model, 1000 bootstrapped data sets, 4 substitution rate categories for across site rate variation, estimated γ distribution parameter, optimized variable sites and empirical nucleotide equilibrium frequencies, a heuristic search of starter tree with BioNJ algorithms, and tree topology search with NNIs. The phylogenetic tree was visualized using FAST Tree program [31].

### 2.8. Annotation, Pan-Genome Analysis, and Genomic Resistance Profile

The sequence was annotated through RAST (Rapid Annotation using Subsystem Technology) database (http://rast.nmpdr.org/, accessed on 10 December 2020). The 32 species RefSeq genomes used for ANI determination were downloaded from NCBI and were annotated through Prokka 1.14.0 [32]. The genome annotation files GFF3 format were used for pan-genome analysis using Pirate [33]. The genetic source of the resistance was evaluated using comparative functions embedded within the PATRIC pipeline and a Comprehensive Antibiotic Resistance Database (CARD, http://arpcard.mcmaster.ca, accessed 28 December 2020) [34]. In addition, Island Viewer was used to identifying genomic islands (GEIs) containing the resistance genes [35]. For the pangenome figures, a new analysis was performed using Anvi’o [36] using the meta-pangenomic workflow with the standard parameters. Genes in the pan-genome were annotated using NCBI’s Clusters of Orthologous Groups.

## 3. Results

### 3.1. Diversity of Culturable Resistant Bacteria from Mine Tailing Contained Soil

A total of 21 bacteria strains were isolated from soil samples belonging to two genera: 18 *Bacillus* (from Firmicutes phylum) and three *Mucilaginibacter* (from Bacteroidetes phylum). All the strains showed resistance to Mn and Co and other multiple resistance to metals and antibiotics (Table 1 and Appendix A). Proteobacteria, Firmicutes, and Bacteroidetes are the most common phyla in the contaminated environment [37]. Despite Proteobacteria being a frequent prevailing phylum, encompassing more than 40% of all prokaryotic genera [38], no strain of this phyla was isolated in the present work. This could be related to the selection pressure exerted by Mn and other metals in the community [39]. In fact, a study in Cu-mine in Brazil reported that Firmicutes was the main isolated phyla [40], a similar response that occurred in the present study.

Based on 16S rDNA analyses, we identified six species of *Bacillus*: *B. aerophilous*, *B. pumilus*, *B. licheniformis*, *B. subtilis*, *B. aryabhattai*, *B. megaterium*. The most common ones were *B. subtilis*, *B. megaterium* and *B. pumilus*. *Bacillus* is a genus characterized as aerobic endospore-forming bacteria, known to resist stressing environments like heat, radiation. The genus also participates in biogeochemical cycling of metals, facilitating oxidation/reduction processes [41]. Observing the phylogenetic tree (Appendix A), some clusters of *Bacillus* (6p, 11p, 12p, 17p, 18p) are less clustered in related to others. One possible cause of this isolation could be related to specialization due to the high metal concentration present in this environment.

All *Bacillus* species isolated in this study have been described with multiple-metal and antibiotic resistance, mainly isolated from the wastewater treatment, or associated with plant growth-promoting. Among them, *B. subtilis* and *B. licheniformis* are the most investigated, with high resistance of oxidative stress and with bioprospection application [42]. Specifically for Mn resistance, strains of *B. subtilis* were described to be involved in manganese oxidation, having an important role in precipitating Mn [43]. Other species also have strains already described with such potential, including as growth-promoters. *B. aerophilous* (strain TR15c), with multiple-metal tolerance including Cu (1750 mg kg^−1^), antibiotic resistance (ampicillin, kanamycin, chloramphenicol, penicillin, tetracycline, and streptomycin), and plant growth-promoting attributes (phosphate solubilization and indole-3-acetic acid production) [44]. *B. aryabhattai*, strain AB211, was described utilizing the root exudates and other organic materials as an energy source, has genes for heat and cold shock antibiotic/metal resistance that enable bacteria to survive biotic/abiotic stress [45] *B. megaterium* (MNSH1-9K-1) showed an elevated resistance to potentially toxic elements and possesses the ability to remove metals like Ni, Hg, and V from liquid media [46].

The content of soil metals where the strain was isolated was as follows: 30 g kg^−1^ for Fe and 460, 3.1, 9, 30, 10.7, 10.1, and 48 mg·kg^−1^ for Mn, Cd, Co, Cu, Ni, and Zn respectively. Despite the results below the threshold value according to Brazilian legislation [47] soil metals content is 3–50 times higher compared to values from the soil before the disaster [48] (Appendix A). Studies in the same contained area reported a high concentration of As, Cd, Pb, Zn, and Mn in sediments and water, causing cumulative effects in the muscle’s fishes after the disaster [5,49]. Queiroz et al. (2021) highlighted an increase of 880% in the concentration of dissolved Mn [50]. This could explain the presence of a wide bride resistance profile and the numerous resistance gene, mainly to Mn.

We found two isolates affiliated to the *Mucilaginibacter* genus, with the high resistance profile, where we highlighted the isolate 21p, that presented resistance to almost all the tested metal and antibiotics. *Mucilaginibacter* is a recent genus, was described for the first time in 2007, as a member of the Bacteroidetes phylum [9]. Bacteroidetes is known for multiply resistance due to natural resistance to aminoglycosides as well as resistance that is acquired during horizontal gene transference (HGT) [51]. The genus is specialized in the degradation of complex organic matter and is known due to the production of large amounts of extracellular polymeric substances (EPS) [9]. EPS was already described as: sorption of metals, growth promotor, and osmoprotector in plants roots [52]. Some strains of this genus are known for their resistance to metals, including Mn since strains of *Mucilaginibacter* were already isolates from Mn-mines presented high levels of metal(loids) resistance [52,53]. Recent studies have highlighted the genera as a root endophytic bacterium in wide broad varieties of plants, promoting growth in stress environments (e.g., salty, water restriction, metal-contaminated) [52,54,55].

In general, both genera isolated in our survey, *Bacillus* and *Mucilaginibacter*, have bioremediation potential. However, *Bacillus* has been more studied with multiple uses, including the production of polymeric compounds and biosolvants, a well-known for sorption and complex pollutants [56]. *Mucilaginibacter*, on the other hand, is a new genus with a low number of identified species with a potentiality that is still not well explored, which supports our decision to investigate more about bacteria from this genus in the present work (Table 1). All of them were resistance to Mn and Co to the maximum concentration and just a few for Cd and Zn. Resistance of NA was rarely found. Isolate 21p were the colony that presented higher resistance among metals and antibiotic and were selected for the futures steps.

### 3.2. A potential Novel Metal(loid) Resistant Species from Mucilaginibacter Genus

*Mucilaginibacter* sp. 21p was addressed by the sequencing of the complete circular genome with one contig with 4,739,655 bp. The genomic GC content of strain 21p, directly calculated from its genome sequence, was determined to be 43.2% which is in the range of the previously described species of the genus *Mucilaginibacter*, from 39.1 to 47.8%. It was annotated 4334 coding sequencing in RAST and 45.47% CDS in PATRIC (RAST tool kit) [9]. Average nucleotide identity (ANI) presents low similarity with all 60 *Mucilaginiobacter* genomes from GenBank, inferior to 95%—proposed threshold for species identity [57]. The highest identity was obtained with *Mucilaginibacter* sp. MYSH2 (accession GCA_003432115) with 80.08%, isolated from beach sand in Yanfyand, South Korea (Appendix A). The lowest identity was with *M. ginsenosidivorans* (3017T) with only 70.22%, isolated from soil of ginseng field, Pocheon province, South Korea [58]. Based on the species Reference Genomes (Appendix A), the strain with up to 74.1% identity was obtained with *M. rigui*, a strain isolated from a freshwater from a wetland, also in South Korea [59] and lowest, we have the same results, being *M. ginsenosidivorans* the most distance species from the genus. These results suggest that our strain 21p belongs to genus *Mucilaginibacter*, and that might be representative of a new species.

A phylogenetic tree of 16S rDNA including representative sequences supports the affiliation of this strain at *Mucilaginibacter* genus (Appendix A), as can be seen in a well-defined genus clade with maximum support of the branch. The similarity between strain 21p and other members of the genus *Mucilaginibacter* was also supported by the phylogenetic tree’s topology of the complete genome (Figure 2). The 21p showed higher stronger lineage (100% bootstrap value) with seven described species besides *M. rigui* as showed by ANI evidence; those include *M. pedocola*, *M. pychrotolerans*, *M. pankratovii*, *M. glaciei*, *M. phyllosphaerae*, *M. rigui*, *M. terrigena*. Despite the closer distance of 21p with those species, our assembly seems to be arranged in a separate independent clade, suggesting that it might have gone through recent speciation events. And among the non-described genomes, the stronger lineage with 21p ones is *M.* sp. MYSH2, described above, *M.* sp. 44–25, and *M.* sp. MD40. For instance, altogether, this genomic evidence strongly suggests that the 21p strain could be a new member of the *Mucilaginibacter* genus.

Species of this genus has been isolated from a wide range of terrestrial and aquatic habitats. The genus *Mucilaginibacter* has a strictly aerobic or facultatively anaerobic metabolism, variable for catalase and oxidase, and high activities of extracellular polymeric substances [10]. This versatile metabolic characteristic allows the genus to be distributed in a wide ecological niche, having a key role in degrading various biopolymers as *M. gynuensis* isolated from wood [60] and oxidating metal as *M. rubeus* and *M. kameinonensis* isolated from contaminated soil [61].

Among the seven closest described species at the phylogenetic tree, *M. pedocola* was the only isolated in metal contaminated areas. *M. psychrotolerant*, *M. glaciei*, *M. pankratovii*, *M. rigui* were isolated in flesh water or soils with high moisture, as peatland, similar areas than 21p were isolated (Figure 3). For this reason, the elicitation of the mechanism of adaption of this strain could be important for the understanding of the impact of metal contamination in the sampled environment.

In order to characterize the functional genomic variation of the 21p strain we conducted a pangenome analysis for the determination of the orthologous gene family’s presence/absence across 32 representative *Mucilaginibacter* species. Pangenome results revealed the presence of gene coding resistance in our assembly that is shared with just one or two other species as *yojl* and *cusC*, related with efflux plumps, important for the resistance of bacteria (Table 2) and the absence of others genes (Appendix A).

Our assembly 21p also has unique alleles, as *dltC*, *rpoE_4*, *menE*, *mmgB*, *rfbD* and *glmE* and some of them are located in GI, suggesting that were horizontally transferred (Table 2). Bastiat (2012) described *rpoE_4* as an extra-cytoplasmic function sigma factor, activated during the generation of sulfite compounds (thiosulfate and taurine) that controls a response required for efficient growth in the presence of sulfite [62] (Table 2). The same authors, also suggesting that it may be advantageous for bacteria in the stationary phase by providing either a sulfite detoxification function or an energy input through sulfite respiration [62]. The other unique resistance gene is *glmE*, related to cobalamin production (vitamin B12)—a cobalt-containing tetrapyrrole cofactor involved in intramolecular rearrangement reactions [63]. Recently some authors have been suggested cobalamin contribute to maintaining the redox balance under highly oxidizing conditions, providing a specific advantage in extremely acidic and highly metal-loaded environments by increasing the tolerance and fitness of these microorganisms [63]. Both unique genes seem to be compatible with the Fe and Mn contaminated environment where 21p were isolated.

Other unique genes integrated important operons that seem to be incomplete in 21p genome as: *srf* responsible for surfactin biosynthesis *srfAA*, *srfAB*, missing *srfAC*; *menE*, for melaquinone synthesis, missing *menB*, *menC*; and *rfbC*, *dTDP-*Rhamnose synthesis, missing *rfbA*. The loss of this genes along generations should be investigated, mainly surfactin biosynthesis operon since it has an important role in bioremediation. Some of the unique genes, as *rpoE_4* and *splG*, are located among 2.55–2.36 M, area associated with one of eleven Genomic Island (GI) harbored in the genome (Figure 3B). Typically, GEIs have been shown to be associated with host-beneficial adaptive traits such as bioremediation, virulence, antibiotic resistance, and metabolism [35,64] (Figure 3B).

Among the unique genes of the three closest strains, there is *splG* gene that encodes a thermophilic spore photoproduct lyase. This lyase belongs to a family of radical S-adenosylmethionine (AdoMet) enzymes [65]. This enzyme is normally present in species, from *Bacillus* and *Clostridium* genus, that are extremely resistant to harsh (physical, chemical, and biological) conditions allowing them to survive. Since 21p was resistant to a high level of metals and antibiotics, the presence of *splG* collaborated with the resistance profile [65]. Other genes related to metal resistant as *arsC*, *acr3*, *mntH*, *corA*, *zraR*, *rpoE*, are present in the other strains, integrating the core genome.

### 3.3. Antibiotic Resistance Genetic Profile of Mucilaginibacter 21p

To gain insight into the genetic basis of how the strain was able to deal with high antibiotic concentrations, we study the annotated genes. Draft genomes were automatically annotated through the RAST database by PATRIC. Putative resistance genes of six antibiotic resistance family were found in our assembly: quinolone (*gyrA*), aminoglycoside (*katG*, *gidB*, *str*), β-lactamase (*bLc*, *ampG*, *ampH*, *pse2*, *ybxl*, *CS*), sulfonamide (*folP*, *folA*), tetracycline (*s10p*, *tet*), daptomycin (*gdpD*), and other (*pxyR*, *kasA*, *ef-tu*, *Iso-tRNA*, *ddl*, *dxr*, *fab*l) (Table 2). Fluoroquinolone and tetracycline resistance was a strict match two classes antibiotic for resistance gene database (CARD).

Some mechanisms of resistance could not be restricted just to antibiotics. When a single mechanism confers resistance to both an antibiotic and metal is called cross-resistance. The *mdtABC*, a multiple drug transporter, is a plasmid-encoded efflux pump that confers resistance for a high concentration of metals (zinc and copper) and β-lactam antibiotic [66,67]. In *czcABC*—operon that confers resistance to zinc, cadmium, and cobalt. This pump efflux, when co-regulatory with gene *czcR*, also present in 21p can confer resistance carbapenems, a class of last resort antibiotic, by repressing the expression of the oprD porin the route of entry for these antibiotics to the bacterial cell [68].

The mechanism as cross-resistance is responsible for bacteria co-selection, which could maintain and promote antibiotic resistance in indigenous bacterial populations even in the absence of antibiotics [69,70]. This topic is a public health concern since soil bacteria could transfer ARG to pathogenic bacteria of clinic relevance [71,72]. Is already known that the exposition of multiple metal increase antibiotic resistance horizontal transference [73]. Exposure to multiple metals, as happens in this contaminated area, has been shown to be more effective for co-selection for antibiotic resistance than exposure to a single metal [74,75]. Recent studies also described those mechanisms of cross-resistance are related to the sub-lethal concentration of metals. So that, this led us understand that antibiotic resistance in 21p could be a consequence of the rich metal(loid) environment where the strain was isolated (Appendix A).

Most of the metal(loid) resistance is supported by biosorption, without energy cost process, so that, to defend themselves from metal toxicity, microbes have evolved a lot of mechanisms of adaption and resistance. Essentially, the diverse determinants maintaining metal(loid) homeostasis can be divided into four categories: pump efflux (ATPase, RND, CDF family), enzymatic detoxification (redox and (de)methylation), intracellular sequestration, and reduction of uptake. Other unspecific mechanisms enhancing bacterial resistance to potential toxic elements are extracellular polymers (EPS) or siderophores secreted by bacteria to trap metal(loid)s, reducing its bioavailability and further alleviating stress [76,77].

The strain 21p isolated presented two major mechanisms of resistance: efflux pump, a mechanism that helps the microorganism to extrude antibiotics and disinfectants and EPS in an efficient manner. On its genome, it is observed genes related with five families of efflux pumps: (i) major facilitator superfamily (MFS); (ii) the ATP-binding cassette superfamily (ABC transport); (iii) the small multidrug resistance (SMR) family; (iv) the resistance-nodulation-division (RND) superfamily; and (v) the multidrug and toxic compound extrusion (MATE) family [78]. Our assembly has at least one representative mechanism from each family, which can explain the wide metal resistance (Table 2).

Studies in the same estuarine area reported a high concentration of As, Cd, Pb, Zn, Mn in sediments and water, causing cumulative effects in the muscle’s fishes after the disaster [5,50,79,80]. Queiroz et al. (2021) highlighted an increase of 880% in the concentration of dissolved Mn. 21p has resistance genes to all these five metals cited in the studies: As (*arsC*, *arsM*, *Acr3*), Zn (*zra*, *zraR*, *Xref* regulator Zn sigma dependent, putative zinc protease, Zn resistance associates’ protein), Mn (*mneP* and manganese efflux: *mntR*, *mnH*, *mnmACEG*); Co (*corA*, cobalt/magnesium transport, and the operon *czc* (*czcA*, *czcB*, *czcD*, *czcR*), cobalt, zinc, and cadmium). We can observe that Mn resistance is the one that presented more diversity of genes related to efflux, (operon *mnm* + *mntR*, *mnH*), what can be correlating to Mn be the metal(loid) with the highest concentration.

Other species from the genus isolated in soils with high concentrations of metal(loids) *M. pedocola*, *M. rubeus*, *M. kameinonensis*, do not present such a wide resistance profile. The resistance is multicopper oxidases, *ars* operons, RND transport systems (*czc*, *cus*, *ncc*). Bacteria exposed at metal(loid) can develop resistance through mutation, but mainly to horizontal transference since requires specific multigene resistances. HT is promoted by multiple metal(loid) contamination and biofilm formation, conditions that occurred here related to the recent evolution of the genus.

The other way to resist metals is promoted by using extracellular polymeric substances (EPS). The genes related to EPS are formation, as mannose, flippase, lipopolysaccharide formation and export system, and glycosyltransferase (Excell EPS- Vide Notion EPS) are abundant in the 21p genome. EPS was one most important characteristics of the genus, being even inference in the name. Bioprospection application of EPS in the genus is already described for (i) bioremediation, as metal(loid) removal, and (ii) growth promoter in extreme environments [81,82].

Microbial EPS binds metal(loids) in sites present in the cellular polymeric structure, complexing metals and micro-precipitated them, without the involvement of energy [83]. Strain TBZ30T from the genus, *M. Pedocola*, adsorb nearly 60% of Zn^2+^ and 55% of Cd^2+^, the medium (added with 0.3 mM ZnSO_4_ and 0.25 mM CdCl_2_, respectively), which is possibly intermediated by the production of EPS [84].

*M. gossypii* sp. and *M. gossypiicola* sp. were described as growth promoter in cotton plants [85]. Some field studies the genus is highlighted as growth promoting in different plants [86], alleviating salt stress [84], water deficiency soil [87] and metal(loid) contaminated environments [53]. The EPS effect is not related to auxin hormone but osmoprotectants changing ion balance in the soil environment and nutrient solubilizing ability [88,89]. Biofilm formation in the genus should be better understand, probably the genus count with an alternative to chemical signaling, as electrical, since no quorum sensing genes have been described in the genus [90,91,92].

## 4. Conclusions

We successfully isolated multiple resistant bacteria from mine tailing contaminated soils in the Rio Doce estuary. The bacterial genera obtained were *Bacillus* and *Mucilaginibacter.* We depicted the genome of the most resistant bacteria obtained. Our results suggest that this strain is possible a new species of the genus. A high number of genes related efflux plump with metal(loid) and antibiotic resistance and EPS production was found. This study describes multiple resistance genes of the 21p of and provides the better understanding of the genes related efflux plump with metal(loid) and antibiotic resistance and EPS production. Our assembly presented some genomic island with rare and unique genes among the genus, what could suggest a recent speciation process due to the high concentration of metal in the soil. A further experimental investigation is required to described *Mucilaginibacter* sp. 21p as a new species and explore the functionality of these strain in bioremediation and/or growth promoter in high metal(loid) contaminated soil. New studies should be considered to better understand the potential of this specie on bioremediation process and the capacity of horizontal transference of these resistances.

## Figures and Tables

**Figure 1 genes-13-00174-f001:**
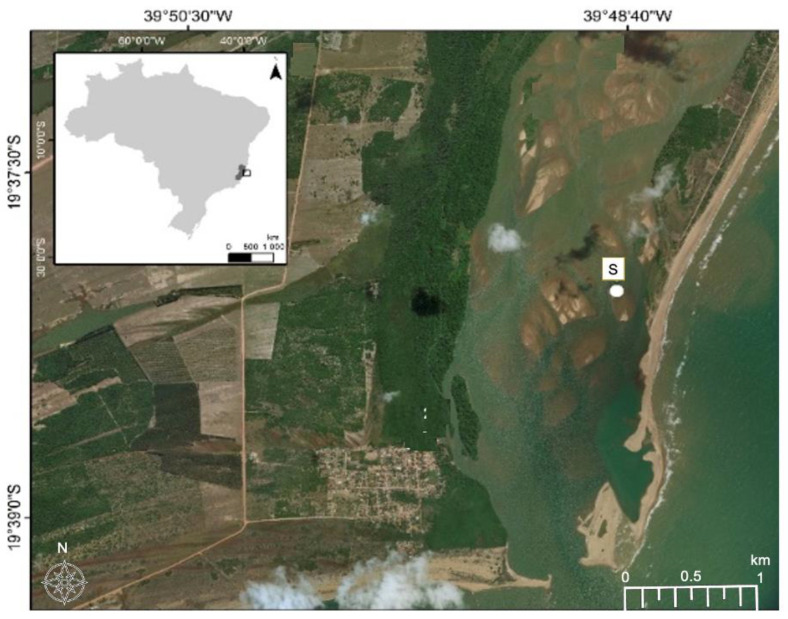
Image of the Rio Doce estuary indicating the locations of sampling sites (S). Coordinate System. GCS WGS 1984 Datum: WGS 1984.

**Figure 2 genes-13-00174-f002:**
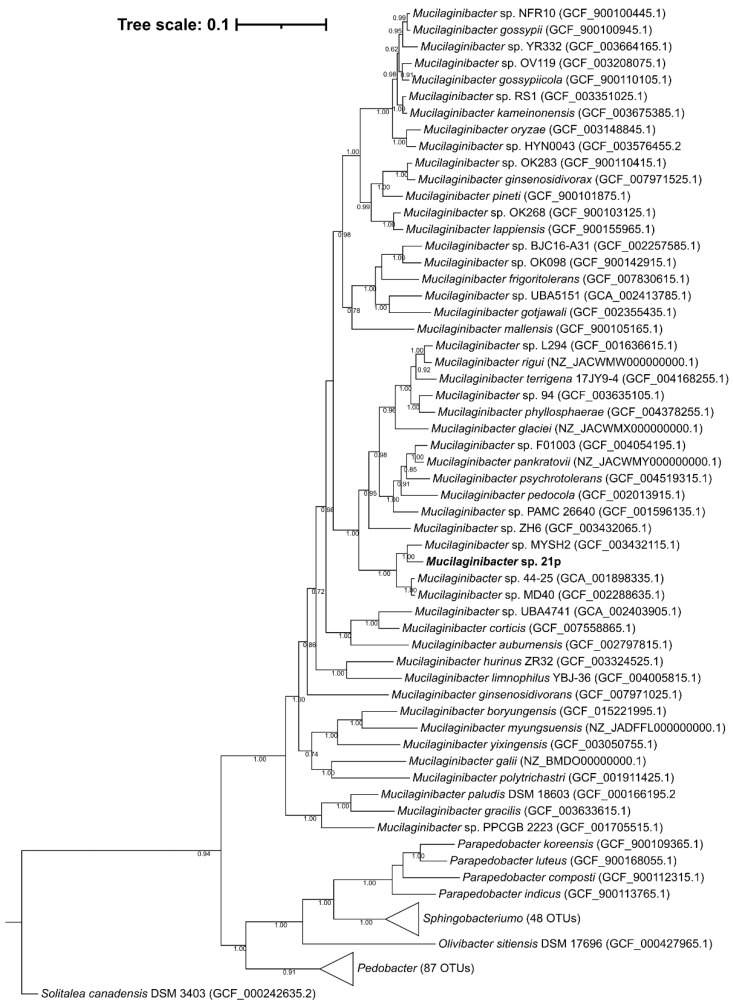
Phylogenetic positioning of the isolate *Mucilaginibacter* sp. 21p (highlighted in bold) based on the maximum likelihood phylogenomic tree based on 120 single-copy conserved amino acid sequences. GenBank Assembly accession code in parentheses. Percentages of bootstraps values above 50% are presented. Bar: 0.1 substitutions per position.

**Figure 3 genes-13-00174-f003:**
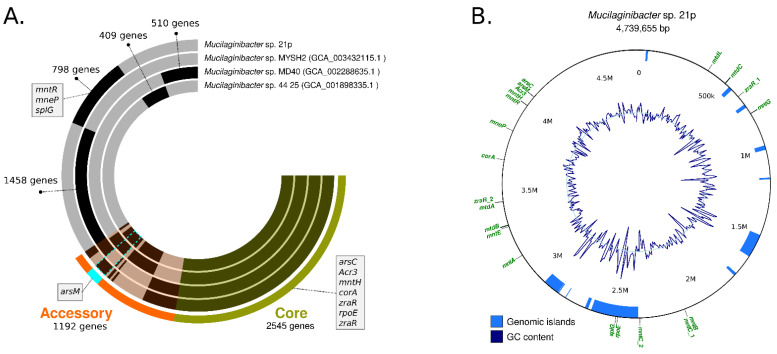
(**A**) Circular visualization of the genomes from *Mucilaginibacter* sp. 21p and the three closest strains according to the phylogenomic tree. Genomes are represented in each radial layer with the black bars representing clusters of genes. Environmental resistance genes present among clustered gene calls are indicated; (**B**) Circular visualization of the genomic island of *Mucilaginibacter* sp. 21p, GC content, and some resistance genes distributed in the genome.

**Table 1 genes-13-00174-t001:** Resistance profile of environmental isolates from contaminated soil and colonies characteristics.

Isolate	Metal				Antimicrobial						Characteristic	
	Resistance Profile (mmol/L)				Minimum Inhibitory Concentration (µg/mL)						Cell/Colony	
	Cd^2+^	Zn^2+^	Co^2+^	Mn^2+^	Amp	Cl	St	NA	Tet	Ka	Gram	Color
1p		500	800	3200	200		100			100	+	White
2p			800	3200						100	+	White
3p	150	500	800	3200		70	100		30		+	White
4p			800	3200						100	+	White
5p	150	500	800	3200	200						+	White
6p			800	3200						100	+	White
7p			800	3200							+	White
8p			800	3200							+	White
9p	30	500	800	3200	10	50	100		30		+	White
10p			800	3200	300	10					+	White
11p			800	3200						100	+	White
12p			800	3200							+	White
13p		200	800	3200	300	10	100	200			+	White
14p			800	3200						100	+	White
15p			800	3200							+	White
16p			800	3200							+	White
17p			800	3200							+	White
18p		500	800	3200		10					+	White
19p			800	3200							−	Pink
20p	300	10	800	3200	300	10	50	50	10		−	Pink
21p	300	10	800	3200	300	10	100	100	10	50	−	Pink

Amp = Ampicicline, Cl = Chloramphenicol; St = Streptomycin, NA = Nalitric Acid, Tet = Tetracycline, Ka = Kanamicyn.

**Table 2 genes-13-00174-t002:** List of rare or unique genes of *M. 21p* according to Pirate.

Gene *	Description	Frequency	Other Species with the Gene	GI (*Loci*)
*cotSA*	Spore coat protein cotAS	rare	*M. ginsenosidivorans*	
*dltC*	D-alanyl carrier protein	unique		
*endOF2*	Endo-β-N-acetylglucosaminidase F2	rare	*M. lappiensis*	
*cusC*	Cation efflux system protein	rare	*M. gossypiicola*	
*fabG*	3-oxoacyl-[acyl-carrier-protein] reductase	rare	*M. ginsenosidivorans*	
*glmE*	Glutamate mutase epsilon subunit	unique		
*lgrB*	Linear gramicidin synthase subunit B	rare	*M. rubeus*	
*lgrE*	Linear gramicidin dehydrogenase	rare		
*menE*	2-succinylbenzoate--CoA ligase	unique		
*mmgB*	putative 3-hydroxybutyryl-CoA dehydrogenase	unique		
*rcsB*	Transcriptional regulatory protein	rare	*M. gossypii*; *M. corticis*	Yes (2,368,463)
*rfbD*	UDP-galactopyranose mutase	unique		
*rpoE_4*	ECF RNA polymerase sigma-E factor	unique		Yes (2,473,906)
*splG*	Spore photoproduct lyase	unique		Yes (2,484,144)
*srfAA*	Surfactin synthase subunit 1	unique		
*srfAB*	Surfactin synthase subunit 2	unique		
*tmoS*	Sensor histidine kinase	rare	*M. pineti*; *M. polytrichastri*	Yes (2,596,785)
*yojI*	ABC transporter ATP-binding/permease protein	rare	*M. lappiensis*; *M. oryzae*	

* All hypothesis genes were excluded from the list. Comparisons were made with the 32 species available at NCBI database at the time of the analysis were made.

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
