# Peer review of "Mucilaginibacter sp. Strain Metal(loid) and Antibiotic Resistance Isolated from Estuarine Soil Contaminated Mine Tailing from the Fundão Dam"

_genes, 2022, doi:10.3390/genes13020174_

Round 1
Reviewer 1 Report
The submitted manuscript studied the metalloid resistance of Mucilaginibacter sp. strain, isolated from estuarine soil contaminated by mine tailing from the Fundao Dam in Brazil. The authors aimed to isolate and identify indigenous bacteria with resistance to metal(loid) that can also increase antibiotic resistance. The Introduction is concise but clear and provides sufficient insight into the issue, including the rationale for the study. The description of Material and Methods is adequate to repeat the experiments. Results and data interpretation are straightforward and lead to reasonable conclusions suitably described in the Discussion and Conclusions. However, the order and numbering of the results in the tables and plots seem chaotic. For example, in Appendix, the authors start with Figure S1, continue with Table S2, Figure S3, Table S4, etc. Both tables and graphs should be numbered separately. Moreover, some tables and plots mentioned in the text are missing or possess the wrong number (e.g. Table S3, S5, S8, Tables S4 and S9 are the same). Some are not mentioned in the text (e.g. Table S11-S13). It must be improved because it degrades otherwise quality work.
My comments:
Line 113: Hours instead of horas.
Line 170: The first sentence, “The text continues here”, should be removed.
Figures 3b and S5 are unreadable.
Author Response
Reviewer 1– Comment#1: “The submitted manuscript studied the metalloid resistance of Mucilaginibacter sp. strain, isolated from estuarine soil contaminated by mine tailing from the Fundão Dam in Brazil. The authors aimed to isolate and identify indigenous bacteria with resistance to metal(loid) that can also increase antibiotic resistance. The Introduction is concise but clear and provides sufficient insight into the issue, including the rationale for the study. The description of Material and Methods is adequate to repeat the experiments. Results and data interpretation are straightforward and lead to reasonable conclusions suitably described in the Discussion and Conclusions. However, the order and numbering of the results in the tables and plots seem chaotic. For example, in Appendix, the authors start with Figure S1, continue with Table S2, Figure S3, Table S4, etc. Both tables and graphs should be numbered separately. Moreover, some tables and plots mentioned in the text are missing or possess the wrong number (e.g.: Table S3, S5, S8, Tables S4 and S9 are the same). Some are not mentioned in the text (e.g.: Table S11-S13). It must be improved because it degrades otherwise quality work.”
Authors: We appreciated the comments. We organized the numbers as suggested.
Reviewer 1– Comment#2: “Line 113: Hours instead of horas.”
Authors: We corrected as suggested.
Reviewer 1– Comment#3: “Line 170: The first sentence, “The text continues here”, should be removed.”
Authors: Excluded as suggested.
Reviewer 1– Comment#3: Figures 3b and S5 are unreadable.
Authors: The figure was adjusted or removed.

Reviewer 2 Report
Dear editor,
I red carrefully the article entitled :"Mucilaginibacter sp. strain metal(loid) resistance isolated from estuarine soil contaminated mine tailing from the Fundão Dam."which is proposed by the authors for publication in Genes Journal. Although several interesting data are presented here, slight changes and minor corrections are required. Please find my comments in this following part.
This paper describes the isolation and genetic characterization of a bacterial strain named Mucilaginibacter sp. from metal and metalloid contaminated soil, following the 2015 breach of a mine dike releasing millions of tons of sediment in an estuary located in southwest Brazil. The first step of this work consisted in selecting, through an original cultivable approach, bacterial strains capable of growing on a rich medium diluted to 10% and supplemented with solutions of antibiotics and various metals & metalloids. This step allowed selecting 21 isolates, which were characterized by 16S metabarcoding and tested for their antibiotic resistance. Two isolates showed resistance properties towards different metals & metalloids and antibiotics; one isolate belonging to the genus Bacillus and one isolate affiliated to the genus Mucilaginibacter. Since few studies are available in the literature concerning the genus Mucilaginibacter and since the isolate named "21p" presented interesting potentialities with capacity to metal-metalloid & antibiotic co-resistance, the authors retained this strain for the continuation of the work. In a second part, the authors performed the complete sequencing of the genome of the isolate "21p" as well as a fine annotation in order to try to identify resistance genes related to the potential of the bacterial strain, accompanied by a phylogenetic characterization from data of the bacterial genus Mucilaginibacter.
This work presents originality, in that it focuses on the description of a bacterial genus poorly documented and, in particular, the characterization of an isolate capable of co-resisting six antibiotics (ampicillin, chloramphenicol, streptomycin, natrilic acid, tetracycline and kanamycin) and cadmium, zinc, cobalt and manganese and likely to belong to a species not yet described in the literature.
Here are the main key points that remain to be completed or improved:
- Modify the title of the article. A possible title might be: “Mucilaginibacter sp: a new metal(loid) and antibio-resistance strain isolated from estuarine soil contaminated mine from the Fundao Dam.
- Add the name of the author corresponding to the research structure N°5.
- All genus and species of the bacterial strains presented in the text should be written in italics. Please correct them.
*Materials & Methods section:
-The part concerning the antibiotic resistance evaluation protocol must be described more precisely here. In particular, the procedure for obtaining MICs for all the antibiotics tested should be described in a more consistent manner so that this part echoes the results presented in Table 1.
- The soil sampling procedure should be specified.
- Table S4 should be completed and added to the Material & Methods section: the information already contained in this table should be completed by providing a full description of the soil structure as well as a physico-chemical analysis describing the main edaphic parameters (pH, organic matter, ...).
- Table S9 should be deleted. It is redundant with the Table S4. Be careful to check the units presented in this table. Are they really kg/m3 ?
- Remove the dot on lines 129-130 between "S1" and "Illumina", on line 145: between "performed" and "based".
- Remove the sentence "The text continues here" line 170.
*Results section:
- Lines 185-187: Change the sentence and complete.
- Lines 205-206: change “mg.kg-1” to “mg.kg-1”
- Table S4 needs to be presented in this section.
- In Table 1, correct the spelling of the following two antibiotic compounds: “Chloramphenicol”, “Streptomycin”.
- Line 241: Add references for this section regarding the Mucilaginibacter genus.
- Insert Table S7 in the results section and not in the appendix.
- Table S7 should be presented in the results section as it presents the intrinsic genetic characteristics of the new strain described by the authors.
- Lines 308-309: Improve this section by expanding and completing with bibliographic references.
- Figure 3 should be enlarged because the text and some of the illustrations are very difficult to read.
- Line 327: In section entitled "antibiotic resistance genetic profile of Mucilaginibacter": Table 1 is not mentioned in the text, please refer to it here. This section needs to be expanded to more fully discuss the antibiotic resistance results of the isolates tested and the MICs in particular.
- Line 375: change "21p" to "21p isolate".
- Line 405: change "inion balance" to "Ion balance".
- Figures S5 and S10 are unreadable as they are. Please modify.
I think this section should be improved in order to provide a glimpse of future work the authors will need to undertake with this new bacterial isolate.
Author Response
Reviewer 2– Comment#1: “I red carefully the article entitled :"Mucilaginibacter sp. strain metal(loid) resistance isolated from estuarine soil contaminated mine tailing from the Fundão Dam."which is proposed by the authors for publication in Genes Journal. Although several interesting data are presented here, slight changes and minor corrections are required. Please find my comments in this following part.
This paper describes the isolation and genetic characterization of a bacterial strain named Mucilaginibacter sp. from metal and metalloid contaminated soil, following the 2015 breach of a mine dike releasing millions of tons of sediment in an estuary located in southwest Brazil. The first step of this work consisted in selecting, through an original cultivable approach, bacterial strains capable of growing on a rich medium diluted to 10% and supplemented with solutions of antibiotics and various metals & metalloids. This step allowed selecting 21 isolates, which were characterized by 16S metabarcoding and tested for their antibiotic resistance. Two isolates showed resistance properties towards different metals & metalloids and antibiotics; one isolate belonging to the genus Bacillus and one isolate affiliated to the genus Mucilaginibacter. Since few studies are available in the literature concerning the genus Mucilaginibacter and since the isolate named "21p" presented interesting potentialities with capacity to metal-metalloid & antibiotic co-resistance, the authors retained this strain for the continuation of the work. In a second part, the authors performed the complete sequencing of the genome of the isolate "21p" as well as a fine annotation in order to try to identify resistance genes related to the potential of the bacterial strain, accompanied by a phylogenetic characterization from data of the bacterial genus Mucilaginibacter.
This work presents originality, in that it focuses on the description of a bacterial genus poorly documented and, in particular, the characterization of an isolate capable of co-resisting six antibiotics (ampicillin, chloramphenicol, streptomycin, natrilic acid, tetracycline and kanamycin) and cadmium, zinc, cobalt and manganese and likely to belong to a species not yet described in the literature.
Reviewer 2– Comment#2: - Modify the title of the article. A possible title might be: “Mucilaginibacter sp: a new metal(loid) and antibio resistance strain isolated from estuarine soil contaminated mine from the Fundao Dam.
Authors: We did not make the microscopy and other tests to confirm that this is a new species. So, the suggestion was partially accepted.
Reviewer 2– Comment#3: Add the name of the author corresponding to the research structure N°5.
Authors: The name was added as suggested.
Reviewer 2– Comment#4: - All genus and species of the bacterial strains presented in the text should be written in italics. Please correct them.
Authors: The name was added as suggested.
Reviewer 2– Comment#5: -The part concerning the antibiotic resistance evaluation protocol must be described more precisely here. In particular, the procedure for obtaining MICs for all the antibiotics tested should be described in a more consistent manner so that this part echoes the results presented in Table 1.
Authors:
We rewrite as suggested:
“Metals and antibiotics solutions were added to the medium after sterilization by filtering using a 0.22 m membrane (Millipore nitrocellulose GSWP 04700) [18]. All the solution were added in warm media avoiding precipitation. The antibiotics and metal were diluted in sterile water, except for chloramphenicol which was diluted in ethanol 70 %. All 21 isolates were inoculated in the plate and the ones that presented minimum grown were considered resistant. After the tests we stored in tryptic soy broth (TSB; Fisher Scientific Inc., Hampton, NH) with 20 % glycerol at -80 ÌŠC.”
Reviewer 2– Comment#6: - The soil sampling procedure should be specified.
Authors: We specified in the line 65 and added more descriptions about area:
“The Rio Doce is a major river in southeastern Brazil. The basin has been modified for several decades of human activities, e.g.: waste discharge, agriculture, aquaculture, industrial and mining activities. In November 2015, a large amount of Fe-enriched mine tailings mostly composed by highly crystalline Fe oxyhydroxides (e.g., hematite and goethite) were dumped into Rio Doce Basin after the Fundão Dam collapse, reaching the Rio Doce Estuary. The presence of tailing was visible in the water and in first 10 cm of the soil. For this study, contaminated soil was collected in December 2018 from the Rio Doce estuary on the southwest coast of Brazil (latitude 19o 23' 28" S, longitude 40o 04' 20" O), 24 months after the disaster, (Figure 1). The soils were sampled using polyvinyl chloride tubes attached to a sampler used flooded soils. After the tubes were hermetically sealed and transported in a vertical position (at approximately 4 °C). We collected the first 5-10 cm depth from the core, where mine tailing was visible and homogenized in Ziplock bag and kept in cold stored chamber (5°C) for 5 days. Soil redox potentials (Eh) were measured during sample collection using a Pt electrode, with the Eh values adjusted to a calomel reference electrode by adding +244 mV. Soil pH values were recorded with a glass electrode, previously calibrated with standard solutions of pH 4.0 and 7.0. A sub-sample was collected for total metal contents analysis, determined by plasma atomic emission spectroscopy (ICP-OES) after microwave-assisted triacid digestion (HCl + HNO3 + HF; USEPA, 1996) (Table 4S).”
Reviewer 2– Comment#7: - Table S4 should be completed and added to the Material & Methods section: the information already contained in this table should be completed by providing a full description of the soil structure as well as a physico-chemical analysis describing the main edaphic parameters (pH, organic matter, ...).
Authors: The information were added in Material and methods and S4.
Reviewer 2– Comment#8 - Table S9 should be deleted. It is redundant with the Table S4. Be careful to check the units presented in this table. Are they really kg/m3?
Authors: We deleted the Table S9 and corrected the table unit.
Reviewer 2– Comment#9: - Remove the dot on lines 129-130 between "S1" and "Illumina", on line 145: between "performed" and "based".
Authors: We deleted the dots among the words.
Reviewer 2– Comment#10: - Remove the sentence "The text continues here" line 170.
Authors: We deleted.
Reviewer 2– Comment#11: - Lines 185-187: Change the sentence and complete.
Authors: We made the modifications
Reviewer 2– Comment#12: - Lines 205-206: change “mg.kg-1” to “mg.kg-1”
Authors: it was correted.
Reviewer 2– Comment#13: - Table S4 needs to be presented in this section.
Authors: We presented in L83.
Reviewer 2– Comment#14: - In Table 1, correct the spelling of the following two antibiotic compounds: “Chloramphenicol”, “Streptomycin”.
Authors: We corrected the words.
Reviewer 2– Comment#15: - Line 241: Add references for this section regarding the Mucilaginibacter genus.
Authors: The reference was included.
Reviewer 2– Comment#16: - Insert Table S7 in the results section and not in the appendix.
Authors: TableS7 was added as Table 2 and was presented in L311.
Reviewer 2– Comment#17: - Table S7 should be presented in the results section as it presents the intrinsic genetic characteristics of the new strain described by the authors.
Authors: We added S7 in the main text and presented as Table 2.
Reviewer 2– Comment#18: - Lines 308-309: Improve this section by expanding and completing with bibliographic references.
Authors: There is not a lot of relative information about the others unique genes, most of them are still under investigation or just doesn’t not bring relevant information.
Reviewer 2– Comment#19: - Figure 3 should be enlarged because the text and some of the illustrations are very difficult to read.
Authors: The picture was modified.
Reviewer 2– Comment#20: - Line 327: In section entitled "antibiotic resistance genetic profile of Mucilaginibacter": Table 1 is not mentioned in the text, please refer to it here. This section needs to be expanded to more fully discuss the antibiotic resistance results of the isolates tested and the MICs in particular.
Authors: We added the description of Table 1 in L251 and added more information about the MIC and the selection of 21p.
“In general, both genera isolated in our survey, Bacillus and Mucilaginibacter, have bioremediation potential. However, Bacillus has been more studied with multiple uses, including the production of polymeric compounds and biosolvants, a well-known for sorption and complex pollutants [58]. Mucilaginibacter, on the other hand, is a new genus with a low number of identified species with a potentiality that is still not well explored, which supports our decision to investigate more about bacteria from this genus in the present work (Table 1). All of them were resistance to Mn and Co to the maximum concentration and just a few for Cd and Zn. Resistance of NA was rarely found. Isolate 21p were the colony that presented higher resistance among metals and antibiotic and were selected for the futures steps.”
Reviewer 2– Comment#21: - Line 375: change "21p" to "21p isolate".
Authors: We modified, following the suggestion
Reviewer 2– Comment#22: - Line 405: change "inion balance" to "Ion balance".
Authors: The word were modified.
Reviewer 2– Comment#23: - Figures S5 and S10 are unreadable as they are. Please modify.
Authors: The figures were modified.
Reviewer 2– Comment#24: I think this section should be improved in order to provide a glimpse of future work the authors will need to undertake with this new bacterial isolate.
Authors: This information was added in the last sentence of the conclusion.
“New studies should be considered to better understand the potential of this specie on bioremediation process and the capacity of horizontal transference of these resistances.”
